# ROBUST CONSTRAINED REINFORCEMENT LEARNING FOR CONTINUOUS CONTROL WITH MODEL MISSPECIFICATION

**Daniel J. Mankowitz**[*]
dmankowitz@google.com

**Dan A. Calian**[*]
dancalian@google.com

**Rae Jeong**      **Cosmin Paduraru**      **Nicolas Heess**      **Sumanth Dathathri**

**Martin Riedmiller**                          **Timothy Mann**

DeepMind
London, UK

## ABSTRACT

Many real-world physical control systems are required to satisfy constraints upon deployment. Furthermore, real-world systems are often subject to effects such as non-stationarity, wear-and-tear, uncalibrated sensors and so on. Such effects effectively perturb the system dynamics and can cause a policy trained successfully in one domain to perform poorly when deployed to a perturbed version of the same domain. This can affect a policy's ability to maximize future rewards as well as the extent to which it satisfies constraints. We refer to this as constrained model misspecification. We present an algorithm that mitigates this form of misspecification, and showcase its performance in multiple simulated Mujoco tasks from the Real World Reinforcement Learning (RWRL) suite.

## 1 INTRODUCTION

Reinforcement Learning (RL) has had a number of recent successes in various application domains which include computer games (Silver et al., 2017; Mnih et al., 2015; Tessler et al., 2017) and robotics (Abdolmaleki et al., 2018a). As RL and deep learning continue to scale, an increasing number of real-world applications may become viable candidates to take advantage of this technology. However, the application of RL to real-world systems is often associated with a number of challenges (Dulac-Arnold et al., 2019; Dulac-Arnold et al., 2020). We will focus on the following two:

**Challenge 1 - Constraint satisfaction**: One such challenge is that many real-world systems have constraints that need to be satisfied upon deployment (i.e., hard constraints); or at least the number of constraint violations as defined by the system need to be reduced as much as possible (i.e., soft-constraints). This is prevalent in applications ranging from physical control systems such as autonomous driving and robotics to user facing applications such as recommender systems.

**Challenge 2 - Model Misspecification (MM)**: Many of these systems suffer from another challenge: model misspecification. We refer to the situation in which an agent is trained in one environment but deployed in a different, perturbed version of the environment as an instance of *model misspecification*. This may occur in many different applications and is well-motivated in the literature (Mankowitz et al., 2018; 2019; Derman et al., 2018; 2019; Iyengar, 2005; Tamar et al., 2014).

There has been much work on constrained optimization in the literature (Altman, 1999; Tessler et al., 2018; Efroni et al., 2020; Achiam et al., 2017; Bohez et al., 2019). However, to our knowledge, the effect of model misspecification on an agent's ability to satisfy constraints at test time has not yet been investigated.

---

[*]indicates equal contribution.

**Constrained Model Misspecification (CMM)**: We consider the scenario in which an agent is required to satisfy constraints at test time but is deployed in an environment that is different from its training environment (i.e., a perturbed version of the training environment). Deployment in a perturbed version of the environment may affect the return achieved by the agent as well as its ability to satisfy the constraints. We refer to this scenario as *constrained model misspecification*.

This problem is prevalent in many real-world applications where constraints need to be satisfied but the environment is subject to state perturbations effects such as wear-and-tear, partial observability etc., the exact nature of which may be unknown at training time. Since such perturbations can significantly impact the agent's ability to satisfy the required constraints it is insufficient to simply ensure that constraints are satisfied in the unperturbed version of the environment. Instead, the presence of unknown environment variations needs to be factored into the training process. One area where such considerations are of particular practical relevance is sim2real transfer where the unknown sim2real gap can make it hard to ensure that constraints will be satisfied on the real system (Andrychowicz et al., 2018; Peng et al., 2018; Wulfmeier et al., 2017; Rastogi et al., 2018; Christiano et al., 2016). Of course, one could address this issue by limiting the capabilities of the system being controlled in order to ensure that constraints are never violated, for instance by limiting the amount of current in an electric motor. Our hope is that our methods can outperform these more blunt techniques, while still ensuring constraint satisfaction in the deployment domain.

**Main Contributions**: In this paper, we aim to bridge the two worlds of model misspecification and constraint satisfaction. We present an RL objective that enables us to optimize a policy that aims to be robust to CMM. Our contributions are as follows: **(1)** Introducing the Robust Return Robust Constraint (R3C) and Robust Constraint (RC) RL objectives that aim to mitigate CMM as defined above. This includes the definition of a Robust Constrained Markov Decision Process (RC-MDP). **(2)** Derive corresponding R3C and RC value functions and Bellman operators. Provide an argument showing that these Bellman operators converge to fixed points. These are implemented in the policy evaluation step of actor-critic R3C algorithms. **(3)** Implement five different R3C and RC algorithmic variants on top of D4PG and DMPO, (two state-of-the-art continuous control RL algorithms). **(4)** Empirically demonstrate the superior performance of our algorithms, compared to various baselines, with respect to mitigating CMM. This is shown consistently across 6 different Mujoco tasks from the Real-World RL (RWRL) suite[1].

## 2 BACKGROUND

### 2.1 MARKOV DECISION PROCESSES

A **Robust Markov Decision Process (R-MDP)** is defined as a tuple $\langle S, A, R, \gamma, \mathcal{P} \rangle$ where $S$ is a finite set of states, $A$ is a finite set of actions, $R : S \times A \to \mathbb{R}$ is a bounded reward function and $\gamma \in [0, 1)$ is the discount factor; $\mathcal{P}(s, a) \subseteq \mathcal{M}(S)$ is an uncertainty set where $\mathcal{M}(S)$ is the set of probability measures over next states $s' \in S$. This is interpreted as an agent selecting a state and action pair, and the next state $s'$ is determined by a conditional measure $p(s'|s, a) \in \mathcal{P}(s, a)$ (Iyengar, 2005). We want the agent to learn a policy $\pi : S \to A$, which is a mapping from states to actions that is robust with respect to this uncertainty set. For the purpose of this paper, we consider deterministic policies, but this can easily be extended to stochastic policies too. The robust value function $V^\pi : S \to \mathbb{R}$ for a policy $\pi$ is defined as $V^\pi(s) = \inf_{p \in \mathcal{P}(s, \pi(s))} V^{\pi, p}(s)$ where $V^{\pi, p}(s) = r(s, \pi(s)) + \gamma p(s'|s, \pi(s)) V^{\pi, p}(s')$. A rectangularity assumption on the uncertainty set (Iyengar, 2005) ensures that "nature" can choose a worst-case transition function independently for every state $s$ and action $a$. This means that during a trajectory, at each timestep, nature can choose any transition model from the uncertainty set to reduce the performance of the agent. A robust policy optimizes for the robust (worst-case) expected return objective: $J_R(\pi) = \inf_{p \in \mathcal{P}} \mathbb{E}^{p, \pi}[\sum_{t=0}^{\infty} \gamma^t r_t]$.

The robust value function can be expanded as $V^\pi(s) = r(s, \pi(s)) + \gamma \inf_{p \in P(s, \pi(s))} \mathbb{E}^p[V^\pi(s')|s, \pi(s)]$. As in (Tamar et al., 2014), we can define an operator $\sigma_{\mathcal{P}(s,a)}^{inf} v : \mathbb{R}^{|S|} \to \mathbb{R}$ as $\sigma_{\mathcal{P}(s,a)}^{inf} v = \inf\{p^\top v | p \in \mathcal{P}(s, a)\}$. We can also define an operator for some policy $\pi$ as $\sigma_\pi^{inf} : \mathbb{R}^{|S|} \to \mathbb{R}^{|S|}$ where $\{\sigma_\pi^{inf} v\}(s) = \sigma_{\mathcal{P}(s, \pi(s))}^{inf} v$. Then, we have defined the Robust Bellman

---

[1]https://github.com/google-research/realworldrl_suite

operator as follows $T_{\mathcal{R}}^{\pi} V^{\pi} = r^{\pi} + \gamma \sigma_{\pi}^{\inf} V^{\pi}$. Both the robust Bellman operator $T_{\mathcal{R}}^{\pi} : \mathcal{R}^{|S|} \rightarrow \mathcal{R}^{|S|}$ for a fixed policy and the optimal robust Bellman operator $T_{\mathcal{R}}^{*} v(s) = \max_{\pi} T_{\mathcal{R}}^{\pi} v(s)$ have previously been shown to be contractions (Iyengar, 2005).

A **Constrained Markov Decision Process (CMDP)** is an extension to an MDP and consists of the tuple $\langle S, A, P, R, C, \gamma \rangle$ where $S, A, R$ and $\gamma$ are defined as in the MDP above and $C : S \times A \rightarrow \mathbb{R}^K$ is a mapping from a state $s$ and action $a$ to a $K$ dimensional vector representing immediate costs relating to $K$ constraint. We use $K=1$ from here on in and therefore $C : S \times A \rightarrow \mathbb{R}$. We refer to the cost for a specific state action tuple $\langle s, a \rangle$ at time $t$ as $c_t(s, a)$. The solution to a CMDP is a policy $\pi : S \rightarrow \Delta_A$ that learns to maximize return and satisfy the constraints. The agent aims to learn a policy that maximizes the expected return objective $J_R^{\pi} = \mathbb{E}[\sum_{t=0}^{\infty} \gamma^t r_t]$ subject to $J_C^{\pi} = \mathbb{E}[\sum_{t=0}^{\infty} \gamma^t c_t] \leq \beta$ where $\beta$ is a pre-defined constraint threshold. A number of approaches (Tessler et al., 2018; Bohez et al., 2019) optimize the Lagrange relaxation of this objective $\min_{\lambda \geq 0} \max_{\theta} J_R^{\pi} - \lambda(J_C^{\pi} - \beta)$ by optimizing the Lagrange multiplier $\lambda$ and the policy parameters $\theta$ using alternating optimization. We also define the constraint value function $V_C^{\pi,p} : S \rightarrow \mathbb{R}$ for a policy $\pi$ as in (Tessler et al., 2018) where $V_C^{\pi,p}(s) = c(s, \pi(s)) + \gamma p(s'|s, \pi(s)) V_C^{\pi,p}(s')$.

## 2.2 CONTINUOUS CONTROL RL ALGORITHMS

We address the CMM problem by modifying two well-known continuous control algorithms by having them optimize the RC and R3C objectives.

The first algorithm is **Maximum A-Posteriori Policy Optimization (MPO)**. This is a continuous control RL algorithm that performs policy iteration using an RL form of expectation maximization (Abdolmaleki et al., 2018a;b). We use the distributional-critic version in Abdolmaleki et al. (2020), which we refer to as DMPO.

The second algorithm is **Distributed Distributional Deterministic Policy Gradient** (D4PG), which is a state-of-the-art actor-critic continuous control RL algorithm with a deterministic policy (Barth-Maron et al., 2018). It is an incremental improvement to DDPG (Lillicrap et al., 2015) with a distributional critic that is learned similarly to distributional MPO.

# 3 ROBUST CONSTRAINED (RC) OPTIMIZATION OBJECTIVE

We begin by defining a Robust Constrained MDP (RC-MDP). This combines an R-MDP and C-MDP to yield the tuple $\langle S, A, R, C, \gamma, \mathcal{P} \rangle$ where all of the variables in the tuple are defined in Section 2. We next define two optimization objectives that optimize the RC-MDP. The first variant attempts to learn a policy that is robust with respect to the return as well as constraint satisfaction - Robust Return Robust Constrained (R3C) objective. The second variant is only robust with respect to constraint satisfaction - Robust Constrained (RC) objective.

Prior to defining these objectives, we add some important definitions.

**Definition 1.** *The robust constrained value function $V_C^{\pi} : S \rightarrow \mathbb{R}$ for a policy $\pi$ is defined as*

$$V_C^{\pi}(s) = \sup_{p \in \mathcal{P}(s, \pi(s))} V_C^{\pi,p}(s) = \sup_{p \in \mathcal{P}(s, \pi(s))} \mathbb{E}^{\pi,p} \left[ \sum_{t=0}^{\infty} \gamma^t c_t \right].$$

This value function represents the worst-case sum of constraint penalties over the course of an episode with respect to the uncertainty set $\mathcal{P}(s, a)$. We can also define an operator $\sigma_{\mathcal{P}(s,a)}^{sup} v : \mathbb{R}^{|S|} \rightarrow \mathbb{R}$ as $\sigma_{\mathcal{P}(s,a)}^{sup} v = \sup\{p^{\top} v | p \in \mathcal{P}(s, a)\}$. In addition, we can define an operator on vectors for some policy $\pi$ as $\sigma_{\pi}^{sup} : \mathbb{R}^{|S|} \rightarrow \mathbb{R}^{|S|}$ where $\{\sigma_{\pi}^{sup} v\}(s) = \sigma_{\mathcal{P}(s,\pi(s))}^{sup} v$. Then, we can defined the Supremum Bellman operator $T_{sup}^{\pi} : \mathcal{R}^{|S|} \rightarrow \mathcal{R}^{|S|}$ as follows $T_{sup}^{\pi} V^{\pi} = r^{\pi} + \gamma \sigma_{\pi}^{sup} V^{\pi}$. Note that this operator is a contraction since we get the same result if we replace $T_{inf}^{\pi}$ with $T_{sup}^{\pi}$ and replace $V$ with $-V$. An alternative derivation of the sup operator contraction has also been derived in the Appendix, Section A.3 for completeness.

### 3.0.1 ROBUST RETURN ROBUST CONSTRAINT (R3C) OBJECTIVE

The R3C objective is defined as:

$$\max_{\pi \in \Pi} \inf_{p \in P} \mathbb{E}^{p,\pi} \left[ \sum_t \gamma^t r(s_t, a_t) \right] \text{ s.t. } \sup_{p' \in \mathcal{P}} \mathbb{E}^{p',\pi} \left[ \sum_t \gamma^t c(s_t, a_t) \right] \leq \beta \qquad (1)$$

Note, a couple of interesting properties about this objective: (1) it focuses on being robust with respect to the return for a pre-defined set of perturbations; (2) the objective also attempts to be robust with respect to the worst case constraint value for the perturbation set. The Lagrange relaxation form of equation 1 is used to define an R3C value function.

**Definition 2** (R3C Value Function). *For a fixed $\lambda$, and using the above-mentioned rectangularity assumption (Iyengar, 2005), the R3C value function for a policy $\pi$ is defined as the concatenation of two value functions $\mathbf{V}^\pi = f(\langle V^\pi, V_C^\pi \rangle) = V^\pi - \lambda V_C^\pi$. This implies that we keep two separate estimates of $V^\pi$ and $V_C^\pi$ and combine them together to yield $\mathbf{V}^\pi$. The constraint threshold $\beta$ term offsets the value function, and has no effect on any policy improvement step[2]. As a result, the dependency on $\beta$ is dropped.*

The next step is to define the R3C Bellman operator. This is presented in Definition 3.

**Definition 3** (R3C Bellman operator). *The R3C Bellman operator is defined as two separate Bellman operators $T_{R3C}^\pi = \langle T_{inf}^\pi, T_{sup}^\pi \rangle$ where $T_{inf}^\pi$ is the robust Bellman operator (Iyengar, 2005) and $T_{sup}^\pi : \mathbb{R}^{|S|} \to \mathbb{R}^{|S|}$ is defined as the* sup *Bellman operator. Based on this definition, applying the R3C Bellman operator to $\mathbf{V}^\pi$ involves applying each of the Bellman operators to their respective value functions. That is, $T_{R3C}^\pi \mathbf{V} = T_{inf}^\pi V - \lambda T_{sup}^\pi V_C$.*

It has been previously shown that $T_{inf}^\pi$ is a contraction with respect to the max norm (Tamar et al., 2014) and therefore converges to a fixed point. We also provided an argument whereby $T_{sup}^\pi$ is a contraction operator in the previous section as well as in Appendix, A.3. These Bellman operators individually ensure that the robust value function $V(s)$ and the constraint value function $V_C(s)$ converge to fixed points. Therefore, $\mathcal{T}_{R3C}^\pi \mathbf{V}$ also converges to a fixed point by construction.

As a result of the above argument, we know that we can apply the R3C Bellman operator in value iteration or policy iteration algorithms in the policy evaluation step. This is achieved in practice by simultaneously learning both the robust value function $V^\pi(s)$ and the constraint value function $V_C^\pi(s)$ and combining these estimates to yield $\mathbf{V}^\pi(s)$.

*It is useful to note that this structure allows for a flexible framework which can define an objective using different combinations of* sup *and* inf *terms, yielding combined Bellman operators that are contraction mappings.* It is also possible to take the mean with respect to the uncertainty set yielding a soft-robust update (Derman et al., 2018; Mankowitz et al., 2019). We do not derive all of the possible combinations of objectives in this paper, but note that the framework provides the flexibility to incorporate each of these objectives. We next define the RC objective.

### 3.0.2 ROBUST CONSTRAINED (RC) OBJECTIVE

The RC objective focuses on being robust with respect to constraint satisfaction and is defined as:

$$\max_{\pi \in \Pi} \mathbb{E}^{\pi,p} \left[ \sum_t \gamma^t r(s_t, a_t) \right] \text{ s.t. } \sup_{p' \in \mathcal{P}} \mathbb{E}^{p',\pi} \left[ \sum_t \gamma^t c(s_t, a_t) \right] < \beta \qquad (2)$$

This objective differs from R3C in that it only focuses on being robust with respect to constraint satisfaction. This is especially useful in domains where perturbations are expected to have a significantly larger effect on constraint satisfaction performance compared to return performance. The corresponding value function is defined as in Definition 2, except by replacing the robust value function in the concatenation with the expected value function $V^{\pi,p}$. The Bellman operator is also similar to Definition 3, where the expected return Bellman operator $T^\pi$ replaces $T_{inf}^\pi$.

---

[2]The $\beta$ term is only used in the Lagrange update in Lemma 1.

### 3.1 LAGRANGE UPDATE

For both objectives, we need to learn a policy that maximizes the return while satisfying the constraint. This involves performing alternating optimization on the Lagrange relaxation of the objective. The optimization procedure alternates between updating the actor/critic parameters and the Lagrange multiplier. For both objectives we have the same gradient update for the Lagrange multiplier:

**Lemma 1** (Lagrange derivative). *The gradient of the Lagrange multiplier* $\lambda$ *is* $\frac{\partial}{\partial \lambda} f = -\left( \sup_{p \in \mathcal{P}} \mathbb{E}^{p,\pi} \left[ \sum_t \gamma^t c(s_t, a_t) \right] - \beta \right)$, *where* $f$ *is the R3C or RC objective loss.*

This is an intuitive update in that the Lagrange multiplier is updated using the worst-case constraint violation estimate. If the worst-case estimate is larger than $\beta$, then the Lagrange multiplier is increased to add more weight to constraint satisfaction and vice versa.

## 4 ROBUST CONSTRAINED POLICY EVALUATION

We now describe how the R3C Bellman operator can be used to perform policy evaluation. This policy evaluation step can be incorporated into any actor-critic algorithm. Instead of optimizing the regular distributional loss (e.g. the C51 loss in Bellemare et al. (2017)), as regular D4PG and DMPO do, we optimize the worst-case distributional loss, which is the distance: $d\left( \mathbf{r_t} + \gamma \mathbf{V}_{\hat{\theta}}^{\pi_k}(s_{t+1}), \mathbf{V}_{\theta}^{\pi_k}(s_t) \right)$,

where $\mathbf{V}_{\theta}^{\pi_k}(s_t) = \inf_{p \in \mathcal{P}(s_t, \pi(s_t))} \left[ V_{\theta}^{\pi_k}(s_{t+1} \sim p(\cdot|s_t, \pi(s_t))) \right] - \lambda \sup_{p' \in \mathcal{P}(s_t, \pi(s_t))} \left[ V_{C,\theta}^{\pi_k}(s_{t+1} \sim p'(\cdot|s_t, \pi(s_t))) \right]$; $\mathcal{P}(s_t, \pi(s_t))$ is an uncertainty set for the current state $s_t$ and action $a_t$; $\pi_k$ is the current network's policy, and $\hat{\theta}$ denotes the target network parameters. The Bellman operators derived in the previous sections are repeatedly applied in this policy evaluation step depending on the optimization objective (e.g., R3C or RC). This would be utilized in the critic updates of D4PG and DMPO. Note that the action value function definition, $\mathbf{Q}_{\theta}^{\pi_k}(s_t, a_t)$, trivially follows.

## 5 EXPERIMENTS

We perform all experiments using domains from the Real-World Reinforcement Learning (RWRL) suite[3], namely `cartpole:{balance, swingup}`, `walker:{stand, walk, run}`, and `quadruped:{walk, run}`. We define a task in our experiments as a 6-tuple $T = \langle \texttt{domain}, \texttt{domain\_variant}, \texttt{constraint}, \texttt{safety\_coeff}, \texttt{threshold}, \texttt{perturbation} \rangle$ whose elements refer to the domain name, the variant for that domain (i.e. RWRL task), the constraint being considered, the safety coefficient value, the constraint threshold and the type of robustness perturbation being applied to the dynamics respectively. An example task would therefore be: $T = \langle \texttt{cartpole}, \texttt{swingup}, \texttt{balance\_velocity}, 0.3, 0.115, \texttt{pole\_length} \rangle$. In total, we have 6 different tasks on which we test our benchmark agents. The full list of tasks can be found in the Appendix, Table 7. The available constraints per domain can be found in the Appendix B.1.

The baselines used in our paper can be seen in Table 1. C-ALG refers to the reward constrained, non-robust algorithms of the variants that we have adapted based on (Tessler et al., 2018; Anonymous, 2020); RC-ALG refers to the robust constraint algorithms corresponding to the Bellman operator $T_{RC}^{\pi}$; R3C-ALG refers to the robust return robust constrained algorithms corresponding to the Bellman operator $T_{R3C}^{\pi}$; SR3C-ALG refers to the soft robust (with respect to return) robust constraint algorithms and R-ALG refers to the robust return algorithms based on Mankowitz et al. (2019).

### 5.1 EXPERIMENTAL SETUP

For each task, the action and observation dimensions are shown in the Appendix, Table 6. The length of an episode is 1000 steps and the upper bound on reward is 1000 (Tassa et al., 2018). All the

---

[3]https://github.com/google-research/realworldrl_suite

| Baseline Algorithm | Variants | Baseline Description |
|---|---|---|
| C-ALG | C-D4PG, C-DMPO | Constraint aware, non-robust. |
| RC-ALG | RC-D4PG, RC-DMPO | Robust constraint. |
| R3C-ALG | R3C-D4PG, R3C-DMPO | Robust return robust constraint. |
| R-ALG | R-D4PG, R-DMPO | Robust return. |
| SR3C-ALG | SR3C-D4PG | Soft robust return, robust constraint. |

Table 1: The baseline algorithms used in this work.

network architectures are the same per algorithm and approximately the same across algorithms in terms of the layers and the number of parameters. A full list of all the network architecture details can be found in the Appendix, Table 4. All runs are averaged across 5 seeds.

**Metrics**: We use three metrics to track overall performance, namely: **return** $R$, **overshoot** $\psi_{\beta,C}$ and **penalized return** $R_{penalized}$. The return is the sum of rewards the agent receives over the course of an episode. The constraint overshoot $\psi_{\beta,C} = \max(0, J_C^\pi - \beta)$ is defined as the clipped difference between the average costs over the course of an episode $J_C^\pi$ and the constraint threshold $\beta$. The penalized return is defined as $R_{penalized} = R - \bar{\lambda}\psi_{\beta,C}$ where $\bar{\lambda} = 1000$ is an evaluation weight and equally trades off return with constraint overshoot $\psi_{\beta,C}$.

**Constraint Experiment Setup**: The *safety coefficient* is a flag in the RWRL suite (Dulac-Arnold et al., 2020) that determines how easy/difficult it is in the environment to violate constraints. The safety coefficient values range from 0.0 (easy to violate constraints) to 1.0 (hard to violate constraints). As such we selected for each task (1) a safety coefficient of 0.3; (2) a particular constraint supported by the RWRL suite and (3) a corresponding constraint threshold $\beta$, which ensures that the agent can find feasible solutions (i.e., satisfy constraints) and solve the task.

**Robustness Experimental Setup:** The robust/soft-robust agents (R3C and RC variants) are trained using a pre-defined uncertainty set consisting of 3 task perturbations (this is based on the results from Mankowitz et al. (2019)). Each perturbation is a different instantiation of the Mujoco environment. The agent is then evaluated on a set of 9 hold-out task perturbations (10 for `quadruped`). For example, if the task is $T = \langle \text{cartpole}, \text{swingup}, \text{balance\_velocity}, 0.3, 0.115, \text{pole\_length} \rangle$, then the agent will have three pre-defined pole length perturbations for training, and evaluate on nine unseen pole lengths, while trying to satisfy the balance velocity constraint.

**Training Procedure**: All agents are always acting on the unperturbed environment. This corresponds to the default environment in the dm_control suite (Tassa et al., 2018) and is referred to in the experiments as the nominal environment. When the agent acts, it generates next state realizations for the nominal environment as well as each of the perturbed environments in the training uncertainty set to generate the tuple $\langle s, a, r, [s', s'_1, s'_2 \cdots s'_N] \rangle$ where N is the number of environments in the training uncertainty set and $s'_i$ is the next state realization corresponding to the $i^{th}$ perturbed training environment. Since the robustness update is incorporated into the policy evaluation stage of each algorithm, the critic loss which corresponds to the TD error in each case is modified as follows: when computing the target, the learner samples a tuple $\langle s, a, r, [s', s'_1, s'_2 \cdots s'_N] \rangle$ from the experience replay. The target action value function for each next state transition $[s', s'_1, s'_2 \cdots s'_N]$ is then computed by taking the inf (robust), average (soft-robust) or the nominal value (non-robust). In each case separate action-value functions are trained for the return $Q(s, a)$ and the constraint $Q_C(s, a)$. These value function estimates then individually return the $mean, \inf, \sup$ value, depending on the technique, and are combined to yield the target to compute $\mathbf{Q}(s, a)$.

The chosen values of the uncertainty set and evaluation set for each domain can be found in Appendix, Table 8. Note that it is common practice to manually select the pre-defined uncertainty set and the unseen test environments. Practitioners often have significant domain knowledge and can utilize this when choosing the uncertainty set (Derman et al., 2019; 2018; Di Castro et al., 2012; Mankowitz et al., 2018; Tamar et al., 2014).

## 5.2 MAIN RESULTS

In the first sub-section we analyze the sensitivity of a fixed constrained policy (trained using C-D4PG) operating in perturbed versions of a given environment. This will help test the hypothesis that perturbing the environment does indeed have an effect on constraint satisfaction as well as on return. In the next sub-section we analyze the performance of the R3C and RC variants with respect to the baseline algorithms.

| Base | Algorithm | $R$ | $R_{penalized}$ | $\max(0, J_C^\pi - \beta)$ |
|------|-----------|-----|-----------------|------------------------------|
| D4PG | C-D4PG | $673.21 \pm 93.04$ | $491.450$ | $0.18 \pm 0.053$ |
|      | R-D4PG | $707.79 \pm 65.00$ | $542.022$ | $0.17 \pm 0.046$ |
|      | **R3C-D4PG** | $\mathbf{734.45 \pm 77.93}$ | $\mathbf{635.246}$ | $\mathbf{0.10 \pm 0.049}$ |
|      | RC-D4PG | $684.30 \pm 83.69$ | $578.598$ | $0.11 \pm 0.050$ |
|      | SR3C-D4PG | $723.11 \pm 84.41$ | $601.016$ | $0.12 \pm 0.038$ |
| DMPO | C-MPO | $598.75 \pm 72.67$ | $411.376$ | $0.19 \pm 0.049$ |
|      | R-MPO | $686.13 \pm 86.53$ | $499.581$ | $0.19 \pm 0.036$ |
|      | **R3C-MPO** | $\mathbf{752.47 \pm 57.10}$ | $\mathbf{652.969}$ | $\mathbf{0.10 \pm 0.040}$ |
|      | RC-MPO | $673.98 \pm 80.91$ | $555.809$ | $0.12 \pm 0.036$ |

Table 2: Performance metrics averaged over all holdout sets for all tasks.

### 5.2.1 FIXED POLICY SENSITIVITY

In order to validate the hypothesis that perturbing the environment affects constraint satisfaction and return, we trained a C-D4PG agent to satisfy constraints across 10 different tasks. In each case, C-D4PG learns to solve the task and satisfy the constraints in expectation. We then perturbed each of the tasks with a supported perturbation and evaluated whether the constraint overshoot increases and the return decreases for the C-D4PG agent. Some example graphs are shown in Figure 1 for the cartpole (left), quadruped (middle) and walker (right) domains. The upper row of graphs contain the return performance (blue curve), the penalized return performance (orange curve) as a function of increased perturbations (x-axis). The vertical red dotted line indicates the nominal model on which the C-D4PG agent was trained. The lower row of graphs contain the constraint overshoot (green curve) as a function of varying perturbations. As seen in the figures, as perturbations increase across each dimension, both the return and penalized return degrades (top row) while the constraint overshoot (bottom row) increases. This provides useful evidence for our hypothesis that constraint satisfaction does indeed suffer as a result of perturbing the environment dynamics. This was consistent among many more settings. The full performance plots can be found in the Appendix, Figures 3, 4 and 5 for cartpole, quadruped and walker respectively.

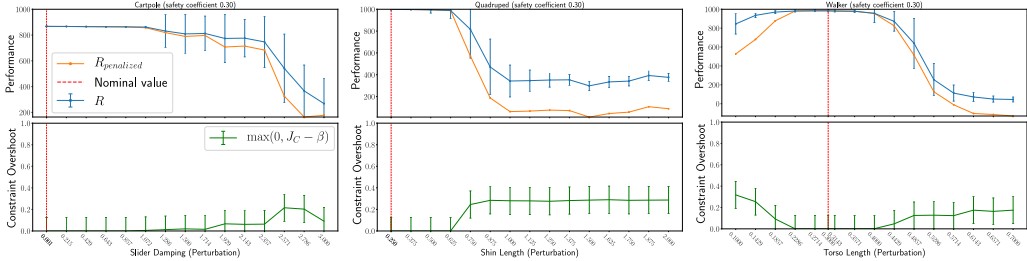

Figure 1: The effect on constraint satisfaction and return as perturbations are added to cartpole, quadruped and walker for a fixed C-D4PG policy.

### 5.2.2 ROBUST CONSTRAINED RESULTS

We now compare C-ALG, RC-ALG, R3C-ALG, R-ALG and SR3C-ALG[4] across 6 tasks. The average performance across holdout sets and tasks is shown in Table 2. As seen in the table, the R3C-ALG variant outperforms all of the baselines in terms of return and constraint overshoot and therefore obtains the highest penalized return performance. Interestingly, the soft-robust variant yields competitive performance.

We further analyze the results for three tasks using ALG=D4PG on the (left column) and ALG=DMPO (right column) in Figure 2. The three tasks are $T_{cartpole, slider\_damping} = \langle \texttt{cartpole}, \texttt{swingup}, \texttt{balance\_velocity}, 0.3, 0.115, \texttt{slider\_damping} \rangle$ (top row), $T_{cartpole, pole\_mass} = \langle \texttt{cartpole}, \texttt{swingup}, \texttt{balance\_velocity}, 0.3, 0.115, \texttt{pole\_mass} \rangle$ (middle row) and $T_{walker} = \langle \texttt{walker}, \texttt{walk}, \texttt{joint\_velocity}, 0.3, 0.1, \texttt{torso\_length} \rangle$ (bottom row). Graphs of the additional tasks can be found in the Appendix, Figures 6 and 7. Each graph contains, on the y-axis, the return $R$ (marked by the transparent colors) and the penalized return $R_{penalized}$ (marked by the dark

---

[4]We only ran the SR3C-D4PG variant to gain intuition as to soft-robust performance.

colors superimposed on top of $R$). The x-axis consists of three holdout set environments in increasing order of difficulty from *Holdout 0* to *Holdout 8*. Holdout N corresponds to perturbation element N for the corresponding task in the Appendix, Table 8. As can be seen for $T_{cartpole,slider\_damping}$ and $T_{cartpole,pole\_mass}$ (Figure 2 (top and middle rows respectively)), R3C-D4PG outperforms the baselines, especially as the perturbations get larger. This can be seen by observing that as the perturbations increase, the penalized return for these techniques is significantly higher than that of the baselines. This implies that the amount of constraint violations is significantly lower for these algorithms resulting in robust constraint satisfaction. $T_{walker}$ (bottom row) has similar performance improved performance over the baseline algorithms.

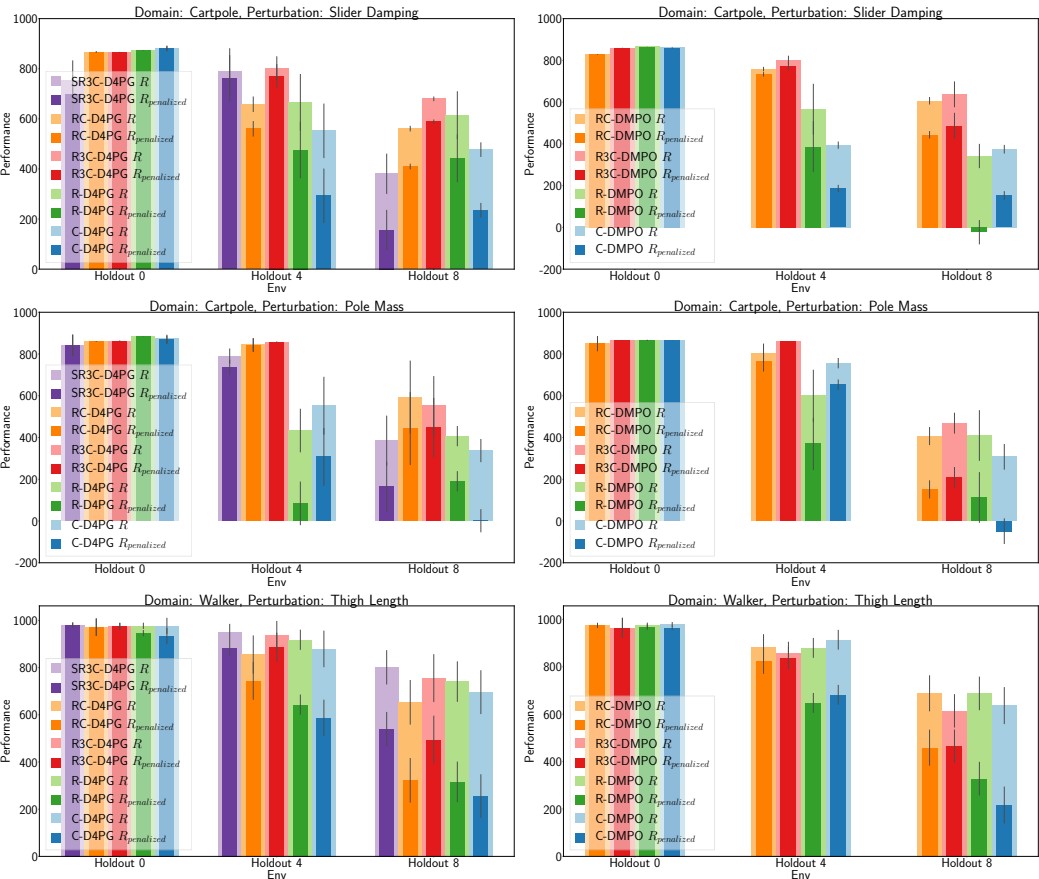

Figure 2: The holdout set performance of the baseline algorithms on D4PG variants (left) and DMPO variants (right) for Cartpole with pole mass perturbations (top row) and walker with thigh length perturbations (bottom row).

## 6 CONCLUSION

This papers simultaneously addresses constraint satisfaction and robustness to state perturbations, two important challenges of real-world reinforcement learning. We present two RL objectives, R3C and RC, that yield robustness to constraints under the presence of state perturbations. We define R3C and RC Bellman operators to ensure that value-based RL algorithms will converge to a fixed point when optimizing these objectives. We then show that when incorporating this into the policy evaluation step of two well-known state-of-the-art continuous control RL algorithms the agent outperforms the baselines on 6 Mujoco tasks. In related work, Everett et al. (2020) considers the problem of being verifiably robust to an adversary that can perturb the state $s' \in S$ to degrade performance as measured by a Q-function. Dathathri et al. (2020) consider the problem of learning agents (in deterministic environments with known dynamics) that satisfy constraints under perturbations to states $s' \in S$. In contrast, equation 1 considers the general problem of learning agents that optimize for the return while satisfying constraints for a given RC-MDP.

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
