# OpenReview forum: "Robust Constrained Reinforcement Learning for Continuous Control with Model Misspecification"
_ICLR.cc/2021/Conference — Reject_

### Official Review · AnonReviewer4 · 2020-10-24
**Encouraging results, but the theory section needs more work**

**Rating:** 4
**Confidence:** 2

**Review:**

The standard Reinforcement Learning framework is limited in many ways, and numerous variants have been introduced to deal with aspects such as partial observability, temporal abstraction, safety, domain transfer, etc. Yet, these issues are often studied separately and it is often unclear how to combine them together. This is the ambitious challenge taken by this paper, which attempts to bridge the two separate settings of Robust MDPs, which aim at considering ambiguity in the dynamics, and Constrained MDPs, which aim at enforcing the satisfaction of a constraint on an expected cost signal. The authors propose the formulation of two objectives, that merge the two aspects and include both a worst-case evaluation over the ambiguity set and a constraint violation penalty term. The ways of dealing with both issues are fairly standard (Lagrangian relaxation of the constraints with alternating optimization, and worst-case evaluation over a finite set of simulated transitions in practice), but their combination seems novel and relevant. These objectives come with the corresponding Bellman Expectation operators, which allow to evaluate the current policy (critic) and provide a feedback (gradient) for the actor to ensure robust constraint satisfaction. The applicability of the proposed approaches is demonstrated on a benchmark of Mujoco tasks, where they are shown to compare favorably to several baselines.

My main concerns lie with the definitions and results of Section 2.3, which I think generally lack rigour and clarity, which sheds doubts on the validity of the claimed results.
1. The authors start by defining the R3V value function $\mathbb{V}$, as a bootstrap of two other values $V$ and $V_c$, that haven't been defined. I was initially confused because they are denoted as if they do not depend on the policy $\pi$, so I first thought these referred to optimal value functions (which would need to be appropriately defined, especially $V_c$ since the costs are constrained rather than optimized), but they seem to be in fact the expected returns for the policy $\pi$ (i.e. the value functions of a policy $\pi$ as opposed to the optimal value functions).
2. Likewise, do the values $V$ and $V_c$ in definition 1 depend on the dynamics $p$? It seems so, but it should be written explicitely.
3. The derivation of A.2 seems a bit sloppy, since the last term in line 4 is identified as$ \mathbb{V}$while it does not strictly correspond to the definition 1.
4. The next state $s'$ is a random variable that depends on the dynamics $p$, and thus subject to the robust inf/sup over $P$ in the objective (1), but in the derivation A.2 and the resulting R3C Bellman operator of definition 2, it is considered as a deterministic variable in which the R3V value can be evaluated freely (without any expectation over $p$, nor inf of $p$ over $P$).
4. In Theorem 1, the R3V values $\mathbb{U}$ and $\mathbb{V}$ are described as functions of $S \to\mathbb{R}^d$, but they were defined as functions of $\to\mathbb{R}$ in definition 1. Also, $d$ is not defined.
5. According to definition 2, the R3V Bellman operator applied to a real function $\mathbb{V}$ simply consists in multiplicating \mathbb{V} by the discount gamma and adding the penalized reward $r - \lambda c$. But then, this is exactly the same as the RC Bellman operator of definition 4. The difference between the two frameworks lies in how the policy value $\mathbb{V}$ is defined (regarding the presence or absence of $\inf_{p\in P}$ before $V^{p,\pi}$), but these differences are not involved when we consider arbitrary functions $\mathbb{V}: S \to\mathbb{R}$ on which to apply the Bellman operators. I feel like  the authors intended the definitions 1 and 3 to be seen somehow as *operators* rather than *functions*, which could allow to retain the sup/inf in the definition of $\mathcal{T}^\pi_{R3C}$ and $\mathcal{T}^\pi_{RC}$, but it is a mere speculation and certainly not what is written in the paper.


In conclusion, this paper comprises a clear motivation, promising insights and encouraging results. But in the present state of vagueness of the theoretical framework, I cannot recommend acceptance. Of course, it may only be a misunderstanding from my part merely related to presentation/clarity issues and not deeper flaws in the reasoning, in which case I am ready to update my rating upon clarification by the authors.

**Minor remarks**:
* Since the definitions and results of Section 2.3 are claimed as novelties, I think they should not be listed as part of the Background section
* After theorem 1, a paragraph states that the $\mathcal{T}_{R3C}$ operator can be used in a Value Iteration algorithm, which is not the case since it is a Bellman expectation operator, not a Bellman optimality operator. If I am not mistaken, it can only be applied in a Policy Iteration scheme.
* The first paragraph of Appendix A3 is self-referencing
* Typo in (20): use norm $||$ instead of absolute value $|$

---

> ### Author Response · Authors · 2020-11-18
> **Thank you for your feedback. We hope we have addressed your concerns.**
>
> Firstly, thank you very much for your in-depth review. We really appreciate your helpful feedback.
>
> “My main concerns lie with the definitions and results of Section 2.3, which I think generally lack rigour and clarity, which sheds doubts on the validity of the claimed results.”
>
> We apologize for the lack of clarity and we appreciate your in-depth comments. The points you made were definitely issues (mostly notational and definitions) that we have now addressed. We have added all of the changes you suggested and fixed the notation in the amended version of the paper. Everything should be much clearer now. Please take a look and let us know if we have missed anything.
>
> “The authors start by defining the R3V value function V, as a bootstrap of two other values V and Vc, that haven't been defined… but they seem to be in fact the expected returns for the policy π (i.e. the value functions of a policy π as opposed to the optimal value functions).”
>
> Thank you for spotting these missing definitions. V is indeed the expected return for the policy pi and V_c is the expected discounted sum of constraint penalties for the policy pi. We have added the definitions to the updated version of the paper.
>
> “Likewise, do the values V and Vc in definition 1 depend on the dynamics p? It seems so, but it should be written explicitely.”
>
> They do depend on the dynamics p. This has also been amended in the updated version.
>
> “The derivation of A.2 seems a bit sloppy, since the last term in line 4 is identified as V while it does not strictly correspond to the definition 1.”
>
> We have made the notation clearer and have introduced an infimum and supremum dynamics operator which should make the relationship clear.
>
> “The next state s′ is a random variable that depends on the dynamics p... it is considered as a deterministic variable in which the R3V value can be evaluated freely ....”
>
> Yes, this is definitely a notational problem. Thank you for pointing this out. We have removed and replaced it with the inf/sup dynamics operators mentioned in the previous answer.
>
> “In Theorem 1, the R3V values U and V are described as functions of S→Rd, but they were defined as functions of →R in definition 1. Also, d is not defined.”
>
> Thank you for spotting this. It has been amended.
>
> “According to definition 2, the R3V Bellman operator applied to a real function V simply consists in multiplicating \mathbb{V} by the discount gamma and adding the penalized reward r−λc... I feel like the authors intended the definitions 1 and 3 to be seen somehow as operators rather than functions, which could allow to retain the sup/inf in the definition of TR3Cπ and TRCπ, but it is a mere speculation and certainly not what is written in the paper.”
>
> This was indeed the intention. This has been modified accordingly to represent that. Is it clearer now?
>
> “Since the definitions and results of Section 2.3 are claimed as novelties, I think they should not be listed as part of the Background section”
>
> Thank you for spotting this. It has been moved into its own section.

---

> > ### Comment · AnonReviewer4 · 2020-11-20
> > **The theory section still needs more work**
> >
> > This revised version is definitely much better than the first one, and I appreciate the efforts of the authors in taking our feedback into account. That being said, there remains a number of concerns in the sections 2 and 3.
> >
> > 1. First, the background section still contains a certain degree of vagueness, and a few typos
> > - Section 2.1: as stated by Reviewer #2, the state space $S$ is not well specified (measurable? finite?). We only have to guess later, from the appearance of $|S|$ in the input space of the operator $\sigma^{inf}_{\mathcal{P}(s,a)}$ that $S$ is assumed to be finite. Yet, neither the analysis nor the experiment seem to rely on this property, it seems that it is only required for the use of the dot-product notation $p^T v$ for than the expectation of $v$ under $p$.
> > - p^\top instead of p^T
> > - typo: we can defined
> > - $\mathcal{R}$ rather than $\mathbb{R}$ used in the sentence "Both the robust Bellman operator...", as noted by Reviewer #3
> > - inconsistent notations: $\mathcal{M}(S)$ for probability measures over states, and $\Delta_A$ for distributions of actions.
> > - From the definition of $\sigma^{inf}_\pi$, it seems that $\pi$ is assumed to be deterministic ($\pi(s)$ notation is used). But 12 lines above $\pi$ was defined as a stochastic policy, mapping to $\Delta_A$. I understand that these notations where borrowed from (Tamar et al., 2014), but in their paper they state that they "shall only consider deterministic policies, and write $\pi(x)$ as the action prescribed by policy $\pi$ at state $x$". But here, this assumption is used without being stated explicitly.
> > - Most previous remarks also apply to Section 3.
> >
> > All these issues are minor and can be fixed in a revised version.
> >
> > 2. Then, we move to Theorem 1 which seems to be claimed as a new result ("we can define the Supremum Bellman Operator, we next prove that this operator is a contraction") whereas it is already known from (Iyengar, 2005): $T^\pi_\sup$ is exactly the definition of the traditional robust Bellman Operator (with a sup instead of inf, which does not change anything, simply replace $V$ by $-V$). Moreover, the definition of $T^\pi_\sup$ as an operator *does not* depend on $V^\pi_C$, which is used as a generic argument, and in turn does not depend on the costs $C$ either. So the name "Sup Constrained Bellman Operator" is misleading, it is only a "Sup Bellman Operator".
> >
> > 3. Similarly, and more importantly, I am still not satisfied with the Definition 3 of the R3C Bellman operator. It is supposed to take as input an R3V Value function $\mathbf{V}: S\to\mathbb{R}$, but the operator can only act on this function as an generic element of $\mathbb{R}^S$. And such an arbitrary function *cannot* be split into two terms $V$ and $V_C$, which would be required to apply the given equation of the operator. To me, this definition is not valid.
> >
> > 4. The first novel result is that of Theorem 2, over the contractivity of $T_{R3C}$. Even though the very definition of $T_{R3C}$ is already concerning as I just said, I also have concerns about the proof provided in Appendix A.4. The authors start by saying that, by definition of the inf operator, for all $\varepsilon$ there exists $p_s\in\mathcal{P}$ such that $E_{p_s}V < \inf_{p\in\mathcal{P}}E_{p}V + \varepsilon$, which is correct. From that, we can derive that $r - \lambda c + \gamma E_{p_s}V -\lambda\gamma E_{p_s}V_C < r -\lambda c + \gamma \inf_{p\in\mathcal{P}} E_{p}V - \lambda\gamma E_{p_s}V_C + \varepsilon$. But then, unless I am mistaken, the term $-\lambda\gamma E_{p_s}V_C$ in the RHS *cannot* be upper-bounded by $-\lambda\gamma\sup_{p'\in\mathcal{P}}{E}_{p'}V_C$, as the authors do. Again, this sheds doubts on the validity of the proof.
> >
> > For this reasoning to be valid, I think that the sup and inf should apply to the same dynamics, i.e. $\inf_p (V - \lambda V_c)$, but that is a different formulation to the one proposed. The authors rejected this idea in their response to Reviewer #3, by saying "This would correspond to learning a worst case reward function and a best case constraint function". Why is that? This objective minimizes the rewards and maximizes the costs, so it is also a worst-case for the constraints.
> >
> > Overall, I still like the original idea of combining the two different safety formulations of RMDPs and CMDPs in the objective (1), which I find novel and interesting, but I maintain my original impression that so far, this paper lacks rigor and clarity in the theory section, which is not suitable for publication.

---

> > > ### Author Response · Authors · 2020-11-20
> > > **Thank you for your reply. We have simplified the arguments and hopefully the paper is clearer.**
> > >
> > > “First, the background section still contains a certain degree of vagueness, and a few typos”
> > >
> > > Thank you for taking another look at the background section. We have incorporated your recommended changes.
> > >
> > > “Then, we move to Theorem 1 which seems to be claimed as a new result”
> > >
> > > We didn’t initially claim this to be a contribution in our introduction, but rather included it for completeness. We have made this much clearer in the paper.
> > >
> > > “Similarly, and more importantly, I am still not satisfied with the Definition 3 of the R3C Bellman operator”
> > >
> > > Thank you for looking through this again. We have decided to significantly modify that section and have removed the theorems as this seems to be a great source of confusion. We now have a much simpler argument as to why the R3C Bellman operator converges to a fixed point. Please let us know if this is clearer.
> > >
> > > “For this reasoning to be valid, I think that the sup and inf should apply to the same dynamics, “
> > >
> > > In the initial reasoning, the theorem assumptions made this a limitation. With the new simpler argument, we can be flexible and the sup and inf terms do not necessarily need to rely on choosing the same transition kernel from the uncertainty set. The uncertainty set, however, still needs to be the same for both the inf and sup terms.
> > >
> > > “"This would correspond to learning a worst case reward function and a best case constraint function". Why is that?”
> > >
> > > You are indeed correct. This however would amount to learning a worst case *combined* value function (and as you mentioned, the same dynamics need to be used for V and V_C). This limits the flexibility of expressing the objective. We may find that a different p from the uncertainty set yields a worse case value for the inf term and a worst case constraint value for the sup term respectively, yielding an even more robust solution. Our simpler argument in the paper makes this possible.
> > >
> > > “Overall, I still like the original idea of combining the two different safety formulations of RMDPs and CMDPs in the objective (1), which I find novel and interesting, but I maintain my original impression that so far, this paper lacks rigor and clarity in the theory section, which is not suitable for publication.”
> > >
> > > We are glad you find this direction interesting. We hope that these changes have provided the paper with more clarity.

---

### Official Review · AnonReviewer2 · 2020-10-29
**An important topic, but the results and the presentation are substandard**

**Rating:** 4
**Confidence:** 4

**Review:**

The paper suggests two approaches to combine the concepts of robust Markov decision processes (MDPs) with that of constrained MDPs. In the first approach, called R3C, a worst-case setting is used for both the expected total discounted rewards criterion and the constraints on the state-action pairs. The robustness is defined with respect to all possible choices (from an uncertainty set) of transition-probability functions. In the second approach, called RC, only the constraints should be robust against all possible transition probabilities. The paper studies the value functions and the corresponding Bellman operators of these problems and argues that, in both cases, these operators are contractions in the supremum norm. Finally, numerical experiments are presented on RWRL problems, such as the cart-pole and the walker, showing the effect of using the redefined operators.

The general problem that the paper studies (namely, robust constrained MDPs) is nice and worth investigation, but the offered combination is straightforward, the theoretical results are weak, and the paper is poorly written (see below). Therefore, the current form of the paper is substandard and needs major improvements.

Several notations and concepts are not specified, starting from the state and action spaces of the MDP. For example, it is not clear whether these spaces are finite, or otherwise, what structure is assumed about them (the minimal assumption that one needs is that they are measurable spaces). The uncertainty set itself is not defined, just the notation is used. The  constraint function $C$ is defined as $S \times A \to \mathbb{R}^K$, but then few lines later in the definition of $J_C^{\pi}$ we simply have $\sum_{t=0}^{\infty} \gamma^t c_t \leq \beta$, without actually stating what $c_t$ is. If $c_t$ is defined as $c_t := C(s_t, a_t)$, then $\beta$ should be a vector to make the above inequality meaningful, with the notation that $\leq$ means coordinate-wise less than or equal. However, things like these should be guessed by the reader as the paper lacks proper definitions.

In Section 2.3.1, about the R3C part, in the second equation after (1), the obtained results are dubious, as ${\bf V}(s')$ should not depend on $s'$, as $s'$ is just a random variable with respect to an expectation is taken (see the definition of the classical Bellman operator). Also function ${\bf V}$ is not defined in the paper, the reader should guess its meaning from the appendix. In section 2.3.1, in equation (2), it is not clear with respect to what probability distribution the expectation is taken in $V(s)$. Is there a special element of the uncertainty set, a "nominal" model?

The structure of the paper is also a bit chaotic. For example, there is an "Experiments" and also an "Experimental Results" part, both containing results of various experiments. Moreover, Sections 6 and 7 should be subsections of Section 5, etc.

---

> ### Author Response · Authors · 2020-11-18
> **Thank you for your feedback. We hope we have addressed your concerns.**
>
> Firstly, thank you very much for your in-depth review. We appreciate your feedback.
>
> “Several notations and concepts are not specified … However, things like these should be guessed by the reader as the paper lacks proper definitions.”
>
> Thank you for your comments.  We had added all the necessary definitions. Let us know if anything is still missing/unclear.
> Note that in our paper we defined the uncertainty set (which is defined previously in literature (e.g., [1]) in the robust MDP background section as follows:
>
> $\mathcal{P}(s,a) \subseteq \mathcal{M}(S)$ is an uncertainty set where $\mathcal{M}(S)$ is the set of probability measures over next states $s' \in S$. This is interpreted as an agent selecting a state and action pair, and the next state $s'$ is determined by a conditional measure $p(s' \vert s, a) \in \mathcal{P}(s,a)$ \citep{iyengar2005robust}.'
> If the reviewer would like something more specific, please let us know.
>
> “In Section 2.3.1, about the R3C part, in the second equation after (1), the obtained results are dubious, as V(s′) should not depend on s′... “
>
> Thank you for pointing this out. This is indeed a notational issue and we have addressed this by adding an infimum and supremum dynamics operator similar to Tamar et. al. (2014). Please see the updated manuscript.
>
> “In section 2.3.1, in equation (2), it is not clear with respect to what probability distribution the expectation is taken in V(s). Is there a special element of the uncertainty set, a "nominal" model?”
>
> Thank you for pointing this out. We have clarified the notation in the updated paper.
>
> “The structure of the paper is also a bit chaotic. For example, there is an "Experiments" and also an "Experimental Results" part, both containing results of various experiments. Moreover, Sections 6 and 7 should be subsections of Section 5, etc.”
>
> Thank you for pointing this out. This indeed was meant to be the case. We have corrected it in the updated manuscript.
>
> * [1] Robust Dynamic Programming, 2005

---

### Official Review · AnonReviewer3 · 2020-10-31
**Recommendation to Reject**

**Rating:** 5
**Confidence:** 4

**Review:**

This manuscript studies the problem of robust and constrained reinforcement learning and proposes two new objectives for incorporating constraints and robustness to misspecified models into RL training. The advantage of these objectives is that their associated Bellman operators are contractive, which enables the use of value-function based methods.

Overall, I vote to reject this manuscript for the reasons detailed below.

Pros:
- Constraints and misspecified models (or models that change over time) are real barriers to deploying RL in many applications. Therefore, the authors study an important problem that has garnered a lot of attention.

- The manuscript offers some theoretical footing and some empirical evidence for the proposed methods.

Cons:
- The discounted penalized cost on the constraints does not enforce the constraints. In my opinion, having a discount factor does not make sense in this case since constraints violations in the future would be discounted. Why wouldn't it be a problem to violate a constraints in the future? I understand that this approach was considered before, but that does not make it a good idea.

- The theoretical results seem to be immediate consequences of the work by Tamar et al, 2014. For example, to show that the sup Bellman operator is a contraction we only need to note that T_sup = - T_inf when we consider the reward -c. Similarly, T_R3C = T_inf when we consider the reward for the T_inf operator to be r - lambda * c. Therefore, all the results follow immediately from Tamar et al. 2014. Given these observations, why is necessary to prove the results in the appendix?

- I do not find the empirical results compelling. I present my concerns in the next bullet points.

- Only three random seeds were used for evaluation which is inadequate for capturing the variance of RL algorithms (see [Henderson et al. "Deep reinforcement learning that matters", 2017] and [Islam et al., "Reproducibility of Benchmarked Deep Reinforcement Learning Tasks for Continuous Control," 2017]). Therefore, comparisons such those presented in Table 2 are not meaningful. Even looking at the values in Table 2 one can see that the mean reward for R3C-D4PG is within a standard deviation of the mean reward of other methods. Similar complaint for the plots shown in Figure 2.

- Although I saw in the appendix the constraint set for each of the tasks considered, I am not sure what the constraint violation costs are. Could the authors clarify this? In particular, I do not know how to interpret the costs shown in the last column of Table 2. How serious is a constraint violation with a cost of 0.113? Also, what are the standard deviations for the constraint costs? Is the difference between 0.113 and 0.128 meaningful?

- Why does it make sense to average performance metrics across tasks in Table 2? Doesn't each task have it's own scale of rewards and costs? I appreciate that there is a breakdown of performance per task in Figure 2.

- Finally, what was the motivation for considering only the MPO and D4PG methods? Would it be possible to try a larger collection of methods on the proposed objectives?

Minor issues:
- All plots are difficult to read.

- Curly R in the first paragraph of section 2.1 is not defined.

- Second line of the second paragraph in section 2.1 states that "C: S x A -> R^K is a K dimensional vector." However, "C: S x A -> R^K" is a map. Also, are multidimensional costs used in the rest of the manuscript? It seemed like all costs were scalars.

- In Definition 1, the values functions V and V_C are never defined (although clear from context).

- The notation \bf{V} (s) = r(s, \pi(s)) + \gamma \bf{V}(s') throughout the paper does not make sense. s' is a random variable whereas \bf{V} (s) and r(s, \pi(s)) are deterministic.

- In the Metrics paragraph, below Table 2, "different" -> "difference"

- Above equation 10 in appendix A.3, the sentence "The full derivation ..." is redundant.

----
Update after rebuttal:

I appreciate the authors' answers and revisions of the manuscript. The theoretical presentation is clearer with the new notation and I appreciate the improved Figure 2.

I appreciate that the authors followed my suggestion to evaluate on more than 3 random seeds. Statistically, however, 5 random seeds are not much different. I was envisioning using at least 30 random seeds. Is computation a major bottleneck? Maybe a simpler and faster method could be used to showcase the benefits of the new objective.

Overall, the new version of the paper is better, but I think the empirical evaluation and the writing should be further improved. I updated my score accordingly.

---

> ### Author Response · Authors · 2020-11-18
> **Thank you for your feedback. We hope that we have addressed your concerns.**
>
> Firstly, thank you very much for your in-depth review. We really appreciate your helpful feedback.
>
> “The discounted penalized cost on the constraints does not enforce the constraints... but that does not make it a good idea.”
>
> A discounted penalized cost means that you have flexibility to: (1) set the discount factor to 1 if you care about constraint violations throughout the course of an episode; (2) set it to less than 1 if you care more about constraint violations early on in an episode. If you look at the real-world RL suite paper [1] (specifically the constrained RL challenge), you will see that in many domains constraint violations occur early on in the episodes. A discounted cost from (2) will focus on mitigating those constraint violations. In addition, as the reviewer mentioned, much previous work has tackled constraints from this perspective (e.g., [2,3,4]).
>
> “The theoretical results seem to be immediate consequences of the work by Tamar et al, 2014. ... T_R3C = T_inf when we consider the reward for the T_inf operator to be r - lambda * c. Therefore, all the results follow immediately from Tamar et al. 2014. Given these observations, why is necessary to prove the results...?”
>
> This is incorrect. T_R3C does not equal T_inf when the reward is r - lambda * c. This would correspond to taking the inf [V - \lambda V_c] which would correspond to learning a worst case reward function and a *best* case constraint function. This would not yield constraint robustness and is not equal to inf(V) s.t. sup(V_C).
> In addition, the results are flexible and extend to all sorts of combinations of Bellman operators. E.g., you can combine inf with the mean or inf with sup etc.
>
> “T_sup = - T_inf when we consider the reward -c.”
>
> We agree. The T_sup = -T_{inf} argument can indeed be made. However, we wanted to explicitly prove this result for reader clarity.
>
> “Only three random seeds were used for evaluation ... Table 2 ... Figure 2.”
>
> We now have 5 seeds for every algorithm variant and have updated the results in the paper.
>
> “... I am not sure what the constraint violation costs are…”
>
> The per-timestep constraint violation costs are 1 if the threshold is violated in a state s and 0 if the state is not violated. This is a standard constraint penalty used in previous works and was also the penalty adopted by the real-world RL suite.
>
> “Why does it make sense to average performance metrics across tasks in Table 2? Doesn't each task have it's own scale of rewards and costs? I appreciate that there is a breakdown of performance per task in Figure 2.”
>
> The aggregate is provided  for ease of comparison. It's meaningful due to the consistent reward scales. All tasks have the same scale of rewards (upper bound of 1000) and costs (between 0 and 1).  As the reviewer mentioned, the breakdown is available as well and consistent with the overall picture.
>
> "Finally, what was the motivation for considering only the MPO and D4PG methods? Would it be possible to try a larger collection of methods on the proposed objectives?"
>
> These are state-of-the-art continuous control RL algorithms. E.g., see [5]. We are also actively trying another technique called Stochastic Value Gradients (SVG) [6]. Our technique can be applied to value based, or actor critic RL algorithms.
>
> "Minor issues:"
>
> “All plots are difficult to read.”
>
> We have made the plots clearer in the paper.
>
> “Curly R in the first paragraph of section 2.1 is not defined.”
>
> We were unable to spot this. Would you mind to please send us the sentence you are referring to?
>
> “Second line of the second paragraph in section 2.1 states that "C: S x A -> R^K is a K dimensional vector." However, "C: S x A -> R^K" is a map. Also, are multidimensional costs used in the rest of the manuscript? It seemed like all costs were scalars.”
>
> Thank you for spotting this. It has been amended.
>
> “In Definition 1, the values functions V and V_C are never defined (although clear from context).”
>
> Thank you for spotting this. We have added definitions of V and V_C to the paper for completeness.
>
> “The notation \bf{V} (s) = r(s, \pi(s)) + \gamma \bf{V}(s') throughout the paper does not make sense. s' is a random variable whereas \bf{V} (s) and r(s, \pi(s)) are deterministic.”
>
> Thank you for this comment. This is a notational issue and we have fixed this in the main draft.
>
> “In the Metrics paragraph, below Table 2, "different" -> "difference"”
>
> Fixed.
>
> “Above equation 10 in appendix A.3, the sentence "The full derivation ..." is redundant.”
>
> Fixed.
>
> * [1] An empirical investigation of the challenges of real-world RL, 2020
> * [2] Reward Constrained Policy Optimization, ICLR, 2018
> * [3] Constrained Policy Optimization, ICML, 2017
> * [4] Constrained Markov Decision Processes via Backward Value Functions, NeurIPS 2020
> * [5] An empirical investigation of the challenges of real-world reinforcement learning (2020)
> * [6] Learning Continuous Control Policies from Stochastic Value Gradients (2015)

---

### Official Review · AnonReviewer1 · 2020-11-03
**Lack of  theoretical contributions and insights**

**Rating:** 5
**Confidence:** 4

**Review:**


In this paper, the author combines robust MDP and constrained MDP for continuous control tasks.

1.First of all, I found such a combination is straightforward. I didn’t see any insight from such combinations

2.The introduction on “constrained model misspecification (CMM)” appears to be unnatural. In control engineering, the perturbation to certain parameters can be treated as an uncertainty; disturbance can appear anytime. And most of the time, such “misspecification” is unknown and can be quantified by some metrics, such as certain norms. Any real-world environment could have such an issue. Many factors need to be considered to get rid of CMM. Therefore, the claimed contribution on “mitigating CMM” is not rigorous and maybe over-claimed.

3.The key contribution is from Theorem1. This is useful to show the algorithm convergence. However, any guarantee of safety and robustness can not be theoretically given, either during training or inference. Theorem 1 does not imply these guarantees.

4.The Lagrangian method is standard to deal with the constrained problem. Theorem 3 is not necessary.

5.The authors should avoid using “real-world” unless real-world experiments are performed. I don’t believe simulations using cart pole, walker, and quadruped are neither enough for illustration nor represent “real-world”, even though the name of the software the authors use is named after “real-world”.

---

> ### Author Response · Authors · 2020-11-18
> **Thank you for your feedback. We hope this has addressed your concerns**
>
> Thank you for taking the time to do the review. We really appreciate your helpful feedback.
>
> “1.First of all, I found such a combination is straightforward. I didn’t see any insight from such combinations”
>
> This approach was born out of a practical need that was not readily available. Current approaches either are (a) robust with respect to return but not constraint satisfaction (e.g., [1]) or (b) simply perform constraint satisfaction without robustness at all (e.g., [2,3]). We agree that our approach is not very complex, but we see this as a plus since less complex methods tend to be adopted more widely.
>
> That being said, we feel that there was a significant amount of insight and novelty to this work, both from a theoretical perspective as well as the thorough empirical analysis that we perform. From a theoretical perspective, it is the first work that proves that combining these two paradigms in the RL setting works with theoretical guarantees. From an empirical perspective, we see that this approach outperforms previous baselines which include variants of (a) and (b) mentioned above. We hope that this work will provide a building block for constrained RL research going forward.
>
> “2.The introduction on “constrained model misspecification (CMM)” appears to be unnatural.”
>
> This approach was born out of a practical need and we have encountered a variety of settings where CMM makes sense. Sim2Real is a nice example where there is clearly a performance gap between the simulation and the real robot. Being robust to the dynamics using (e.g, [4,5]) leads to improved transfer.
>
> “... “mitigating CMM” … maybe over-claimed.”
>
> We certainly agree that other factors can contribute to CMM. However, we tried to be very specific with our claim to mitigating constraint model misspecification. We define the perturbations to the dynamics of the system (where a specific parameter is perturbed. For example, mass, length etc) This is in a continuous control setting and we showcase our results in a Mujoco environment. We mitigate CMM in this setting. If the reviewer has a suggestion as to how we should change our claim, we would be happy to consider it. We also build on the definition of previous works that define model misspecification [e.g., 1] and constrained optimization [e.g., 2].
>
> “3. This is useful to show the algorithm convergence. However, any guarantee of safety and robustness can not be theoretically given, either during training or inference. Theorem 1 does not imply these guarantees.”
>
> Good question. We agree that there is no explicit guarantee on safety and robustness. This is a research direction in and of itself and one that we are actively pursuing. For the given objectives, the guarantee of constraint satisfaction is in expectation (which is a common guarantee for many previous constrained RL works (e.g., [2,3])); as you mentioned, Theorem 1 guarantees that the algorithm will converge.
>
> “The key contribution is from Theorem1.”
>
> Regarding our contributions, we (1) introduce, for the first time in RL, the notion of a robust constrained RL objective; (2) introduce a robust constrained value function; (3) show that the sup Bellman operator for this value function is a contraction; (4) Derive the R3C value function and show that it is a contraction (Theorem 1). In addition, we (5) perform an ablation to verify our hypothesis, (6) implement 9 variants of robust algorithms using DMPO and D4PG and (7) showcase the improved performance on a number of Mujoco domains.
>
> “4.The Lagrangian method is standard to deal with the constrained problem. Theorem 3 is not necessary.”
>
> We agree that the lagrangian method is one of the standard ways to deal with constrained RL. However, we are optimizing for worst-case constraint violations and therefore a supremum term is introduced into the objective and therefore into the gradient update. As such we get a lagrange relaxation with the supremum yielding a conceptually different lagrangian gradient update. We can remove the word theorem and call it a Lemma if that would be better from your perspective?
>
> “5.The authors should avoid using “real-world” ...."
>
> We want to emphasize that our approach is *motivated* by real-world problems. The suite we use is called the real-world RL suite and as such, we use the term real-world when referring to this suite in our paper. We do not claim to have real-world experiments in our paper and have modified some of the narrative to capture this.
>
> * [1] Robust reinforcement learning for continuous control with model misspecification, ICLR, 2020
> * [2] Reward Constrained Policy Optimization, ICLR, 2018
> * [3] Constrained Policy Optimization, ICML, 2017
> * [4] Transfer from simulation to real world through learning deep inverse dynamics model, 2016
> * [5] Learning dexterous in-hand manipulation, 2018

---

### Author Response · Authors · 2020-11-18
**Thank you for your in-depth reviews. Here are the changes made to the manuscript.**

We would like to thank all of the reviewers for their in-depth and helpful reviews. Based on your comments we have (1) added missing definitions, (2) made the notation clearer, (3) improved the overall flow of the paper; (4) we also ran more seeds and have a total of 5 seeds now for both D4PG and DMPO variants; (5) we also have made the figures clearer and easier to parse. We hope our changes have addressed your concerns. We are certainly happier with the overall paper. Please let us know if anything is still not clear.

---

> ### Author Response · Authors · 2020-11-24
> **Thank you for your responses and time. We have one more iteration on the draft.**
>
> We would like to thank all of the reviewers for your in-depth and helpful reviews. The quality of the reviews were excellent and we really feel this has helped shape the paper for future conferences if it does not make it to this one. We did one final iteration on the paper. If you have time, please take a look. If not, we of course understand and appreciate your time taken thus far.

---

### Decision · Program_Chairs · 2021-01-07
**Final Decision**

**Decision:**

Reject

**Comment:**

# Quality:
I personally feel that the comment from Reviewer1 regarding "real-world" is a minor but valid point. Even after the rebuttal, the abstract seems to suggest that the proposed algorithm is effective to solve real-world challenges. Maybe further rephrasing or explicitly stating that the experiments are in simulation might help to clarify this point.

Reviewer3 also raised valid points regarding the experimental evaluation and the use of just 3 seeds. Given the struggle in reproducibility in RL and the shady experimental practices from even leading AI companies (e.g., cherry-picking of seeds), it is paramount that experiments follow strong methodologies and good practices. As such, the experimental results presented in this manuscript should be strengthened.

# Clarity:
All the reviewers pointed out that the paper writing should be improved. Although the authors significantly improved the manuscript during the rebuttal period. Several reviewers suggested that the manuscript should be further polished before publication.

# Originality:
The two proposed approaches are novel to the best of the reviewers and my knowledge. Two reviewers pointed out that the theoretical results should be explained more thoroughly and to clearly differentiate from prior work.

# Significance of this work:
The paper deal with an important and timely topic. Although the work could be very impactful for real-world applications, there is no real-world application. Hence it is difficult to gauge the significance of the work.

# Overall:
The paper does not feel quite ready for publication yet. A clearer presentation and extended experiments would certainly improve the quality of the manuscript.